# Associative Embedding: End-to-End Learning for Joint Detection and Grouping

**Alejandro Newell**
Computer Science and Engineering
University of Michigan
Ann Arbor, MI
alnewell@umich.edu

**Zhiao Huang***
Institute for Interdisciplinary Information Sciences
Tsinghua University
Beijing, China
hza14@mails.tsinghua.edu.cn

**Jia Deng**
Computer Science and Engineering
University of Michigan
Ann Arbor, MI
jiadeng@umich.edu

## Abstract

We introduce associative embedding, a novel method for supervising convolutional neural networks for the task of detection and grouping. A number of computer vision problems can be framed in this manner including multi-person pose estimation, instance segmentation, and multi-object tracking. Usually the grouping of detections is achieved with multi-stage pipelines, instead we propose an approach that teaches a network to simultaneously output detections and group assignments. This technique can be easily integrated into any state-of-the-art network architecture that produces pixel-wise predictions. We show how to apply this method to multi-person pose estimation and report state-of-the-art performance on the MPII and MS-COCO datasets.

## 1 Introduction

Many computer vision tasks can be viewed in the context of detection and grouping: detecting smaller visual units and grouping them into larger structures. For example, in multi-person pose estimation we detect body joints and group them into individual people; in instance segmentation we detect pixels belonging to a semantic class and group them into object instances; in multi-object tracking we detect objects across video frames and group them into tracks. In all of these cases, the output is a variable number of visual units and their assignment into a variable number of visual groups.

Such tasks are often approached with two-stage pipelines that perform detection first and grouping second. But such approaches may be suboptimal because detection and grouping are tightly coupled: for example, in multiperson pose estimation, the same features used to recognize wrists or elbows in an image would also suggest whether a wrist and elbow belong to the same limb.

In this paper we ask whether it is possible to jointly perform detection and grouping using a single-stage deep network trained end-to-end. We propose *associative embedding*, a novel method to express output for joint detection and grouping. The basic idea is to introduce, for each detection, a vector embedding that serves as a "tag" to identify its group assignment. All detections associated with the same tag value belong to the same group. Concretely, the network outputs a heatmap of per-pixel

detection scores and a set of per-pixel embeddings. The detections and groups are decoded by extracting the corresponding embeddings from pixel locations with top detection scores.

To train a network to produce the correct tags, we use a loss function that encourages pairs of tags to have similar values if the corresponding detections belong to the same group or dissimilar values otherwise. It is important to note that we have no "ground truth" tags for the network to predict, because what matters is not the particular tag values, only the differences between them. The network has the freedom to decide on the tag values as long as they agree with the ground truth grouping.

We apply our approach to multiperson pose estimation, an important task for understanding humans in images. Given an input image, multi-person pose estimation seeks to detect each person and localize their body joints. Unlike single-person pose there are no prior assumptions of a person's location or size. Multi-person pose systems must scan the whole image detecting all people and their corresponding keypoints. For this task, we integrate associative embedding with a stacked hourglass network [31], which produces a detection heatmap and a tagging heatmap for each body joint, and then group body joints with similar tags into individual people. Experiments demonstrate that our approach outperforms all recent methods and achieves state-of-the-art results on MS-COCO [27] and MPII Multiperson Pose [3].

Our contributions are two fold: (1) we introduce associative embedding, a new method for single-stage, end-to-end joint detection and grouping. This method is simple and generic; it works with any network architecture that produces pixel-wise prediction; (2) we apply associative embedding to multiperson pose estimation and achieve state-of-the-art results on two standard benchmarks.

## 2 Related Work

**Vector Embeddings** Our method is related to many prior works that use vector embeddings. Works in image retrieval have used vector embeddings to measure similarity between images [12, 43]. Works in image classification, image captioning, and phrase localization have used vector embeddings to connect visual features and text features by mapping them to the same vector space [11, 14, 22]. Works in natural language processing have used vector embeddings to represent the meaning of words, sentences, and paragraphs [30, 24]. Our work differs from these prior works in that we use vector embeddings as identity tags in the context of joint detection and grouping.

**Perceptual Organization** Work in perceptual organization aims to group the pixels of an image into regions, parts, and objects. Perceptual organization encompasses a wide range of tasks of varying complexity from figure-ground segmentation [28] to hierarchical image parsing [15]. Prior works typically use a two stage pipeline [29], detecting basic visual units (patches, superpixels, parts, etc.) first and grouping them second. Common grouping approaches include spectral clustering [41, 36], conditional random fields (e.g. [23]), and generative probabilistic models (e.g. [15]). These grouping approaches all assume pre-detected basic visual units and pre-computed affinity measures between them but differ among themselves in the process of converting affinity measures into groups. In contrast, our approach performs detection and grouping in one stage using a generic network that includes no special design for grouping.

It is worth noting a close connection between our approach to those using spectral clustering. Spectral clustering (e.g. normalized cuts [36]) techniques takes as input pre-computed affinities (such as predicted by a deep network) between visual units and solves a generalized eigenproblem to produce embeddings (one per visual unit) that are similar for visual units with high affinity. Angular Embedding [28, 37] extends spectral clustering by embedding depth ordering as well as grouping. Our approach differs from spectral clustering in that we have no intermediate representation of affinities nor do we solve any eigenproblems. Instead our network directly outputs the final embeddings.

Our approach is also related to the work by Harley et al. on learning dense convolutional embeddings [16], which trains a deep network to produce pixel-wise embeddings for the task of semantic segmentation. Our work differs from theirs in that our network produces not only pixel-wise embeddings but also pixel-wise detection scores. Our novelty lies in the integration of detection and grouping into a single network; to the best of our knowledge such an integration has not been attempted for multiperson human pose estimation.

**Multiperson Pose Estimation** Recent methods have made great progress improving human pose estimation in images in particular for single person pose estimation [40, 38, 42, 31, 8, 5, 32, 4, 9, 13,

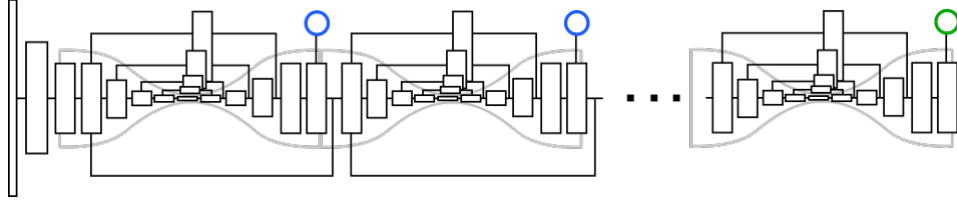

Figure 1: We use the stacked hourglass architecture from Newell et al. [31]. The network performs repeated bottom-up, top-down inference producing a series of intermediate predictions (marked in blue) until the last "hourglass" produces a final result (marked in green). Each box represents a 3x3 convolutional layer. Features are combined across scales by upsampling and performing elementwise addition. The same ground truth is enforced across all predictions made by the network.

26, 18, 7, 39, 34]. For multiperson pose, prior and concurrent work can be categorized as either top-down or bottom-up. Top-down approaches [33, 17, 10] first detect individual people and then estimate each person's pose. Bottom-up approaches [35, 20, 21, 6] instead detect individual body joints and then group them into individuals. Our approach more closely resembles bottom-up approaches but differs in that there is no separation of a detection and grouping stage. The entire prediction is done at once in a single stage. This does away with the need for complicated post-processing steps required by other methods [6, 20].

## 3 Approach

To introduce associative embedding for joint detection and grouping, we first review the basic formulation of visual detection. Many visual tasks involve detection of a set of visual units. These tasks are typically formulated as scoring of a large set of candidates. For example, single-person human pose estimation can be formulated as scoring candidate body joint detections at all possible pixel locations. Object detection can be formulated as scoring candidate bounding boxes at various pixel locations, scales, and aspect ratios.

The idea of associative embedding is to predict an embedding for each candidate in addition to the detection score. The embeddings serve as tags that encode grouping: detections with similar tags should be grouped together. In multiperson pose estimation, body joints with similar tags should be grouped to form a single person. It is important to note that the absolute values of the tags do not matter, only the distances between tags. That is, a network is free to assign arbitrary values to the tags as long as the values are the same for detections belonging to the same group.

To train a network to predict the tags, we enforce a loss that encourages similar tags for detections from the same group and different tags for detections across different groups. Specifically, this tagging loss is enforced on candidate detections that coincide with the ground truth. We compare pairs of detections and define a penalty based on the relative values of the tags and whether the detections should be from the same group.

### 3.1 Network Architecture

Our approach requires that a network produce dense output to define a detection score and vector embedding at each pixel of the input image. In this work we use the stacked hourglass architecture, a model used previously for single-person pose estimation [31]. Each "hourglass" is comprised of a standard set of convolutional and pooling layers to process features down to a low resolution capturing the full global context of the image. These features are upsampled and combined with outputs from higher resolutions until reaching a final output resolution. Stacking multiple hourglasses enables repeated bottom-up and top-down inference to produce a more accurate final prediction. Intermediate predictions are made by the network after each hourglass (Fig. 1). We refer the reader to [31] for more details of the network architecture.

The stacked hourglass model was originally developed for single-person human pose estimation and designed to output a heatmap for each body joint of a target person. The pixel with the highest heatmap activation is used as the predicted location for that joint. The network consolidates global and local features to capture information about the full structure of the body while preserving fine

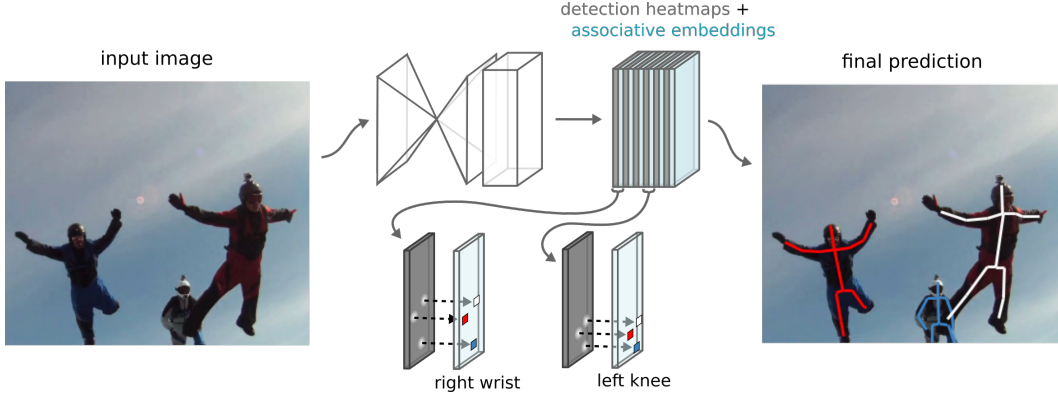

input image

detection heatmaps +
associative embeddings

final prediction

right wrist          left knee

Figure 2: An overview of our approach for producing multi-person pose estimates. For each joint of the body, the network simultaneously produces detection heatmaps and predicts associative embedding tags. We take the top detections for each joint and match them to other detections that share the same embedding tag to produce a final set of individual pose predictions.

details for precise localization. This balance between global and local context is just as important when predicting poses of multiple people.

We make some modifications to the network architecture to increase its capacity and accommodate the increased difficulty of multi-person pose estimation. We increase the number of features at each drop in resolution of the hourglass ($256 \rightarrow 384 \rightarrow 512 \rightarrow 640 \rightarrow 768$). In addition, individual layers are composed of 3x3 convolutions instead of residual modules. Residual links are still included across each hourglass as well as skip connections at each resolution.

## 3.2 Detection and Grouping

For multiperson pose estimation, we train the network to detect joints in a similar manner to prior work on single-person pose estimation [31]. The model predicts a detection score at each pixel location for each body joint ("left wrist", "right shoulder", etc.) regardless of person identity. The difference from single-person pose being that an ideal heatmap for multiple people should have multiple peaks (e.g. to identify multiple left wrists belonging to different people), as opposed to just a single peak for a single target person.

During training, we impose a detection loss on the output heatmaps. The detection loss computes mean square error between each predicted detection heatmap and its "ground truth" heatmap which consists of a 2D gaussian activation at each keypoint location. This loss is the same as the one used by Newell et al. [31].

Given the top activating detections from these heatmaps we need to pull together all joints that belong to the same individual. For this, we turn to the associative embeddings. For each joint of the body, the network produces additional channels to define an embedding vector at every pixel. Note that the dimension of the embeddings is not critical. If a network can successfully predict high-dimensional embeddings to separate the detections into groups, it should also be able to learn to project those high-dimensional embeddings to lower dimensions, as long as there is enough network capacity. In practice we have found that 1D embedding is sufficient for multiperson pose estimation, and higher dimensions do not lead to significant improvement. Thus throughout this paper we assume 1D embeddings.

We think of these 1D embeddings as "tags" indicating which person a detected joint belongs to. Each detection heatmap has its own corresponding tag heatmap, so if there are $m$ body joints to predict then the network will output a total of $2m$ channels; $m$ for detection and $m$ for grouping. To parse detections into individual people, we get the peak detections for each joint and retrieve their corresponding tags at the same pixel location (illustrated in Fig. 2). We then group detections across body parts by comparing the tag values of detections and matching up those that are close enough. A group of detections now forms the pose estimate for a single person.

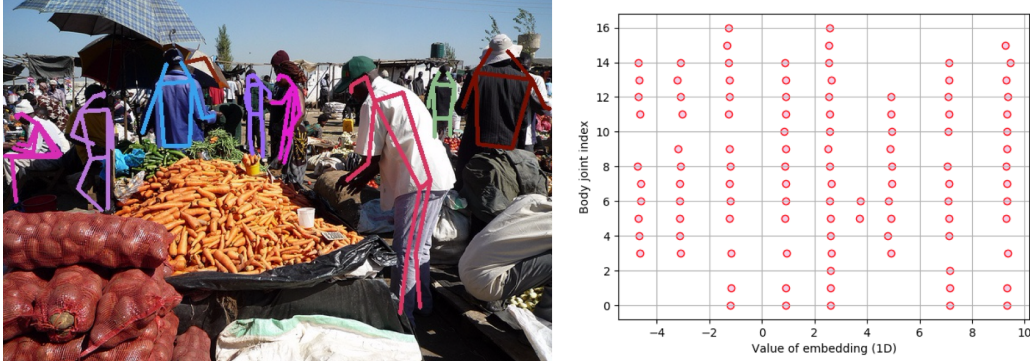

Figure 3: Tags produced by our network on a held-out validation image from the MS-COCO training set. The tag values are already well separated and decoding the groups is straightforward.

The grouping loss assesses how well the predicted tags agree with the ground truth grouping. Specifically, we retrieve the predicted tags for all body joints of all people at their ground truth locations; we then compare the tags within each person and across people. Tags within a person should be the same, while tags across people should be different.

Rather than enforce the loss across all possible pairs of keypoints, we produce a reference embedding for each person. This is done by taking the mean of the output embeddings of all joints belonging to a single person. Within an individual, we compute the squared distance between the reference embedding and the predicted embedding for each joint. Then, between pairs of people, we compare their reference embeddings to each other with a penalty that drops exponentially to zero as the distance between the two tags increases.

Formally, let $h_k \in R^{W \times H}$ be the predicted tagging heatmap for the $k$-th body joint, where $h(x)$ is a tag value at pixel location $x$. Given $N$ people, let the ground truth body joint locations be $T = \{(x_{nk})\}, n = 1, \ldots, N, k = 1 \ldots, K$, where $x_{nk}$ is the ground truth pixel location of the $k$-th body joint of the $n$-th person.

Assuming all K joints are annotated, the reference embedding for the $n$th person would be

$$\bar{h}_n = \frac{1}{K} \sum_k h_k(x_{nk})$$

The grouping loss $L_g$ is then defined as

$$L_g(h, T) = \frac{1}{NK} \sum_n \sum_k \left(\bar{h}_n - h_k(x_{nk})\right)^2 + \frac{1}{N^2} \sum_n \sum_{n'} \exp\{-\frac{1}{2\sigma^2} \left(\bar{h}_n - \bar{h}_{n'}\right)^2\}$$

The first half of the loss pulls together all of the embeddings belonging to an individual, and the second half pushes apart embeddings across people. We use a $\sigma$ value of 1 in our training.

### 3.3 Parsing Network Output

Once the network has been trained, decoding is straightforward. We perform non-maximum suppression on the detection heatmaps and threshold to get a set of detections for each body joint. Then, for each detection we retrieve its corresponding associative embedding tag. To give an impression of the types of tags produced by the network and the trivial nature of grouping we refer to Figure 3; we plot a set of detections where the y-axis indicates the class of body joint and the x-axis the assigned embedding.

To produce a final set of predictions we iterate through each joint one by one. An ordering is determined by first considering joints around the head and torso and gradually moving out to the limbs. We use the detections from the first joint (the neck, for example) to form our initial pool of detected people. Then, given the next joint, say the left shoulder, we have to figure out how to best match its detections to the current pool of people. Each detection is defined by its score and embedding tag, and each person is defined by the mean embedding of their current joints.

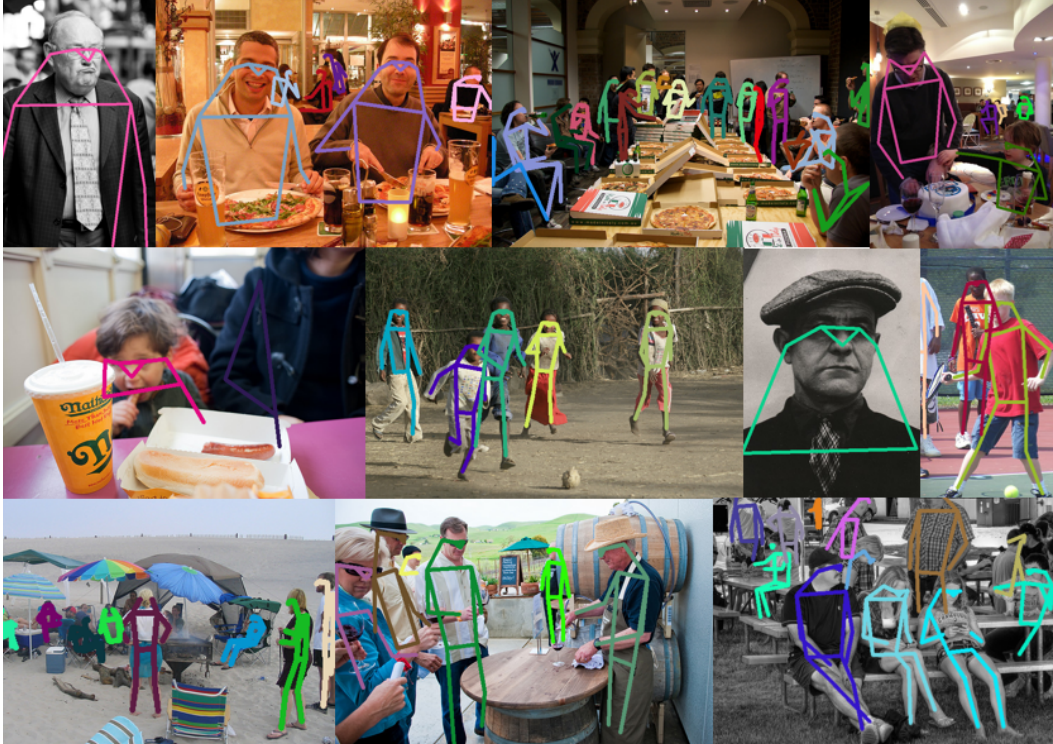

Figure 4: Qualitative results on MSCOCO validation images

We compare the distance between these embeddings, and for each person we greedily assign a new joint based on the detection with the highest score whose embedding falls within some distance threshold. New detections that are not matched are used to start a new person instance. This accounts for cases where perhaps only a leg or hand is visible for a particular person. We repeat this process for each joint of the body until every detection has been assigned to a person. No steps are taken to ensure anatomical correctness or reasonable spatial relationships between pairs of joints.

**Missing joints:** In some evaluation settings we may need to ensure that each person has a prediction for all joints, but our parsing does not guarantee this. Missing joints are usually fine, as in cases with truncation and extreme occlusion, but when it is necessary to produce complete predictions we introduce an additional processing step: given a missing joint, we identify all pixels whose embedding falls close enough to the target person, and choose the pixel location with the highest activation. This score may be lower than our usual cutoff threshold for detections.

**Multiscale Evaluation:** While it is feasible to train a network to predict poses for people of all scales, there are some drawbacks. Extra capacity is required of the network to learn the necessary scale invariance, and the precision of predictions for small people will suffer due to issues of low resolution after pooling. To account for this, we evaluate images at test time at multiple scales. We take the heatmaps produced at each scale and resize and average them together. Then, to combine tags across scales, we concatenate the set of tags at a pixel location into a vector $v \in R^m$ (assuming $m$ scales). The decoding process remains unchanged.

## 4 Experiments

**Datasets** We evaluate on two datasets: MS-COCO [27] and MPII Human Pose [3]. MPII Human Pose consists of about 25k images and contains around 40k total annotated people (three-quarters of which are available for training). Evaluation is performed on MPII Multi-Person, a set of 1758 groups of multiple people taken from the test set as outlined in [35]. The groups for MPII Multi-Person are usually a subset of the total people in a particular image, so some information is provided to make sure predictions are made on the correct targets. This includes a general bounding box and

| | Head | Shoulder | Elbow | Wrist | Hip | Knee | Ankle | Total |
|---|---|---|---|---|---|---|---|---|
| Iqbal&Gall, ECCV16 [21] | 58.4 | 53.9 | 44.5 | 35.0 | 42.2 | 36.7 | 31.1 | 43.1 |
| Insafutdinov et al., ECCV16 [20] | 78.4 | 72.5 | 60.2 | 51.0 | 57.2 | 52.0 | 45.4 | 59.5 |
| Insafutdinov et al., arXiv16a [35] | 89.4 | 84.5 | 70.4 | 59.3 | 68.9 | 62.7 | 54.6 | 70.0 |
| Levinkov et al., CVPR17 [25] | 89.8 | 85.2 | 71.8 | 59.6 | 71.1 | 63.0 | 53.5 | 70.6 |
| Insafutdinov et al., CVPR17 [19] | 88.8 | 87.0 | 75.9 | 64.9 | 74.2 | 68.8 | 60.5 | 74.3 |
| Cao et al., CVPR17 [6] | 91.2 | 87.6 | 77.7 | 66.8 | 75.4 | 68.9 | 61.7 | 75.6 |
| Fang et al., ICCV17 [10] | 88.4 | 86.5 | 78.6 | **70.4** | 74.4 | **73.0** | **65.8** | 76.7 |
| Our method | **92.1** | **89.3** | **78.9** | 69.8 | **76.2** | 71.6 | 64.7 | **77.5** |

Table 1: Results (AP) on MPII Multi-Person.

| | AP | $AP^{50}$ | $AP^{75}$ | $AP^M$ | $AP^L$ | AR | $AR^{50}$ | $AR^{75}$ | $AR^M$ | $AR^L$ |
|---|---|---|---|---|---|---|---|---|---|---|
| CMU-Pose [6] | 0.611 | 0.844 | 0.667 | 0.558 | 0.684 | 0.665 | 0.872 | 0.718 | 0.602 | 0.749 |
| G-RMI [33] | 0.643 | 0.846 | 0.704 | **0.614** | 0.696 | 0.698 | 0.885 | 0.755 | 0.644 | 0.771 |
| Our method | **0.663** | **0.865** | **0.727** | 0.613 | **0.732** | **0.715** | **0.897** | **0.772** | **0.662** | **0.787** |

Table 2: Results on MS-COCO **test-std**, excluding systems trained with external data.

| | AP | $AP^{50}$ | $AP^{75}$ | $AP^M$ | $AP^L$ | AR | $AR^{50}$ | $AR^{75}$ | $AR^M$ | $AR^L$ |
|---|---|---|---|---|---|---|---|---|---|---|
| CMU-Pose [6] | 0.618 | 0.849 | 0.675 | 0.571 | 0.682 | 0.665 | 0.872 | 0.718 | 0.606 | 0.746 |
| Mask-RCNN [17] | 0.627 | **0.870** | 0.684 | 0.574 | 0.711 | – | – | – | – | – |
| G-RMI [33] | 0.649 | 0.855 | 0.713 | **0.623** | 0.700 | 0.697 | 0.887 | 0.755 | 0.644 | 0.771 |
| Our method | **0.655** | 0.868 | **0.723** | 0.606 | **0.726** | **0.702** | **0.895** | **0.760** | **0.646** | **0.781** |

Table 3: Results on MS-COCO **test-dev**, excluding systems trained with external data.

scale term used to indicate the occupied region. No information is provided on the number of people or the scales of individual figures. We use the evaluation metric outlined by Pishchulin et al. [35] calculating average precision of joint detections.

MS-COCO [27] consists of around 60K training images with more than 100K people with annotated keypoints. We report performance on two test sets, a development test set (test-dev) and a standard test set (test-std). We use the official evaluation metric that reports average precision (AP) and average recall (AR) in a manner similar to object detection except that a score based on keypoint distance is used instead of bounding box overlap. We refer the reader to the MS-COCO website for details [1].

**Implementation Details** The network used for this task consists of four stacked hourglass modules, with an input size of $512 \times 512$ and an output resolution of $128 \times 128$. We train the network using a batch size of 32 with a learning rate of 2e-4 (dropped to 1e-5 after about 150k iterations) using Tensorflow [2]. The associative embedding loss is weighted by a factor of 1e-3 relative to the MSE loss of the detection heatmaps. The loss is masked to ignore crowds with sparse annotations. At test time an input image is run at multiple scales; the output detection heatmaps are averaged across scales, and the tags across scales are concatenated into higher dimensional tags.

Following prior work [6], we apply a single-person pose model [31] trained on the same dataset to investigate further refinement of predictions. We run each detected person through the single person model, and average the output with the predictions from our multiperson pose model. From Table 5, it is clear the benefit of this refinement is most pronounced in the single-scale setting on small figures. This suggests output resolution is a limit of performance at a single scale. Using our method for evaluation at multiple scales, the benefits of single person refinement are almost entirely mitigated as illustrated in Tables 4 and 5.

**MPII Results** Average precision results can be seen in Table 1 demonstrating an improvement over state-of-the-art methods in overall AP. Associative embedding proves to be an effective method for teaching the network to group keypoint detections into individual people. It requires no assumptions about the number of people present in the image, and also offers a mechanism for the network to express confusion of joint assignments. For example, if the same joint of two people overlaps at the exact same pixel location, the predicted associative embedding will be a tag somewhere between the respective tags of each person.

We can get a better sense of the associative embedding output with visualizations of the embedding heatmap (Figure 5). We put particular focus on the difference in the predicted embeddings when

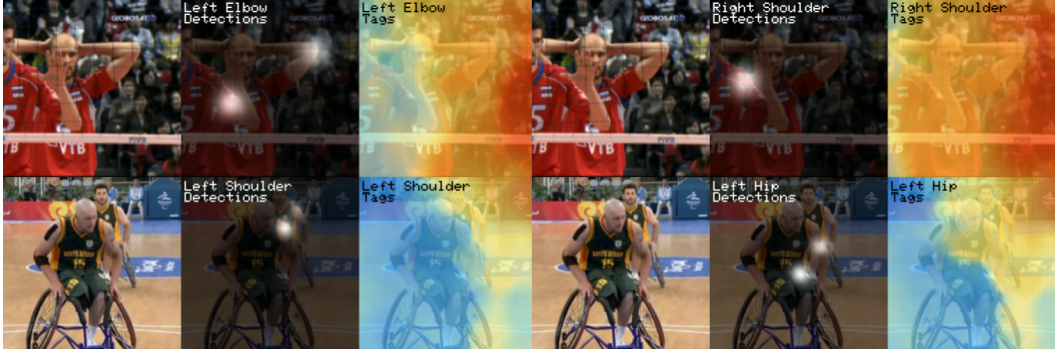

Figure 5: Here we visualize the associative embedding channels for different joints. The change in embedding predictions across joints is particularly apparent in these examples where there is significant overlap of the two target figures.

|  | Head | Shoulder | Elbow | Wrist | Hip | Knee | Ankle | Total |
|---|---|---|---|---|---|---|---|---|
| multi scale | 92.9 | 90.9 | 81.0 | 71.0 | 79.3 | 70.6 | 63.4 | 78.5 |
| multi scale + refine | 93.1 | 90.3 | 81.9 | 72.1 | 80.2 | 72.0 | 67.8 | 79.6 |

Table 4: Effect of single person refinement on a held out validation set on MPII.

|  | AP | $AP^{50}$ | $AP^{75}$ | $AP^M$ | $AP^L$ |
|---|---|---|---|---|---|
| single scale | 0.566 | 0.818 | 0.618 | 0.498 | 0.670 |
| single scale + refine | 0.628 | 0.846 | 0.692 | 0.575 | 0.706 |
| multi scale | 0.650 | 0.867 | 0.713 | 0.597 | 0.725 |
| multi scale + refine | 0.655 | 0.868 | 0.723 | 0.606 | 0.726 |

Table 5: Effect of multi-scale evaluation and single person refinement on MS-COCO **test-dev**.

people overlap heavily as the severe occlusion and close spacing of detected joints make it much more difficult to parse out the poses of individual people.

**MS-COCO Results** Table 2 and 3 report our results on MS-COCO. We report results on both test-std and test-dev because not all recent methods report on test-std. We see that on both sets we achieve the state of the art performance. An illustration of the network's predictions can be seen in Figure 4. Typical failure cases of the network stem from overlapping and occluded joints in cluttered scenes. Table 5 reports performance of ablated versions of our full pipeline, showing the contributions from applying our model at multiple scales and from further refinement using a single-person pose estimator. We see that simply applying our network at multiple scales already achieves competitive performance against prior state of the art methods, demonstrating the effectiveness of our end-to-end joint detection and grouping.

We perform an additional experiment on MS-COCO to gauge the relative difficulty of detection versus grouping, that is, which part is the main bottleneck of our system. We evaluate our system on a held-out set of 500 training images. In this evaluation, we replace the predicted detections with the ground truth detections but still use the predicted tags. Using the ground truth detections improves AP from 59.2 to 94.0. This shows that keypoint detection is the main bottleneck of our system, whereas the network has learned to produce high quality grouping. This fact is also supported by qualitative inspection of the predicted tag values, as shown in Figure 3, from which we can see that the tags are well separated and decoding the grouping is straightforward.

## 5 Conclusion

In this work we introduce associative embeddings to supervise a convolutional neural network such that it can simultaneously generate and group detections. We apply this method to multi-person pose and demonstrate the feasibility of training to achieve state-of-the-art performance. Our method is general enough to be applied to other vision problems as well, for example instance segmentation and multi-object tracking in video. The associative embedding loss can be implemented given any network that produces pixelwise predictions, so it can be easily integrated with other state-of-the-art architectures.

# 6 Acknowledgements

This work is partially supported by the National Science Foundation under Grant No. 1734266. ZH is partially supported by the Institute for Interdisciplinary Information Sciences, Tsinghua University.

## Footnotes

* Work done while a visiting student at the University of Michigan.

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
