[Reviews · NeurIPS 2017]

Reviewer 1



Strengths: - The idea is very novel. Unlike typical approaches to pose estimation that first detect the people and estimate the keypoints, this approach detects keypoints bottom-up, and then groups together keypoints that belong to the same person. The idea of tags to do this grouping is very intuitive, straightforward but effective. - To the best of my knowledge, the final performance is state-of-the-art on all the important benchmarks. - The exposition is clear and understandable. Weaknesses - The model seems to really require the final refinement step to achieve state-of-the-art performance. - How does the size of the model (in terms of depth or number of parameters) compare to competing approaches? The authors mention that the model consists of 4 hourglass modules, but do not say how big each hourglass module is. - There are some implementation details that are curious and will benefit from some intuition: for example, lines 158-160: why not just impose a pairwise relationship across all pairs of keypoints? the concept of anchor joints seems needlessly complex.

Reviewer 2



Paper summary: The paper addresses the problem of detecting multiple people and their corresponding keypoints/joints in images. The paper is an extension of [31], where the neural network predicts an ID for each keypoint as well. Paper strengths: - The proposed method achieves state-of-the-art performance on MPII and COCO (person category) datasets. Paper weaknesses: - The paper is incremental and does not have much technical substance. It just adds a new loss to [31]. - "Embedding" is an overloaded word for a scalar value that represents object ID. - The model of [31] is used in a post-processing stage to refine the detection. Ideally, the proposed model should be end-to-end without any post-processing. - Keypoint detection results should be included in the experiments section. - Sometimes the predicted tag value might be in the range of tag values for two or more nearby people, how is it determined to which person the keypoint belongs? - Line 168: It is mentioned that the anchor point changes if the neck is occluded. This makes training noisy since the distances for most examples are computed with respect to the neck. Overall assessment: I am on the fence for this paper. The paper achieves state-of-the-art performance, but it is incremental and does not have much technical substance. Furthermore, the main improvement comes from running [31] in a post-processing stage.

Reviewer 3



The paper proposes a method for the task of multi-person pose estimation. It is based on the stacked hour glasses architecture for single person pose estimation which is combined with an associative embedding. Body joint heatmaps are associated with tag heatmaps indicating which individual the joint belongs to. A new grouping loss is defined to assess how well predicted tags agree with ground truth joint associations. The approach is evaluated on the MPII multi-person and on MS-COCO, achieving state of the art performance for both of them. Strengths: - State-of-the-art performance is achieved for the multi-person pose estimation problem on MPII Multi-person and MS-COCO. Weakness: - The paper is rather incremental with respect to [31]. The authors adapt the existing architecture for the multi-person case producing identity/tag heatmaps with the joint heatmaps. - Some explanations are unclear and rather vague. Especially, the solution for the multi-scale case (end of Section 3) and the pose refinement used in section 4.4 / table 4. This is important as most of the improvement with respect to state of the art methods seems to come from these 2 elements of the pipeline as indicated in Table 4. Comments: The state-of-the-art performance in multi-person pose estimation is a strong point. However, I find that there is too little novelty in the paper with respect to the stacked hour glasses paper and that explanations are not always clear. What seems to be the key elements to outperform other competing methods, namely the scale-invariance aspect and the pose refinement stage, are not well explained.